# Integrative Transcriptomic and Proteomic Analyses of Molecular Mechanism Responding to Salt Stress during Seed Germination in Hulless Barley

**DOI:** 10.3390/ijms21010359

**Published:** 2020-01-06

**Authors:** Yong Lai, Dangquan Zhang, Jinmin Wang, Juncheng Wang, Panrong Ren, Lirong Yao, Erjing Si, Yuhua Kong, Huajun Wang

**Affiliations:** 1College of Forestry, Henan Agricultural University, Zhengzhou 450002, China; xlaiyong@163.com (Y.L.); zhangdangquan@163.com (D.Z.); 2College of Agriculture and Animal Husbandry, Qinghai University, Xining 810016, China; 3Gansu Provincial Key Lab of Aridland Crop Science, Lanzhou 730070, China; 4Gansu Key Lab of Crop Improvement and Germplasm Enhancement, Lanzhou 730070, China; 5State Key Laboratory of Plant Genomics, National Centre for Plant Gene Research, Institute of Genetics and Developmental Biology, Chinese Academy of Sciences, Beijing 100101, China

**Keywords:** salt stress, seed germination, transcriptome, proteome, hulless barley

## Abstract

Hulless barley (*Hordeum vulgare* L. var. *nudum*) is one of the most important crops in the Qinghai-Tibet Plateau. Soil salinity seriously affects its cultivation. To investigate the mechanism of salt stress response during seed germination, two contrasting hulless barley genotypes were selected to first investigate the molecular mechanism of seed salinity response during the germination stage using RNA-sequencing and isobaric tags for relative and absolute quantitation technologies. Compared to the salt-sensitive landrace lk621, the salt-tolerant one lk573 germinated normally under salt stress. The changes in hormone contents also differed between lk621 and lk573. In lk573, 1597 differentially expressed genes (DEGs) and 171 differentially expressed proteins (DEPs) were specifically detected at 4 h after salt stress, and correspondingly, 2748 and 328 specifically detected at 16 h. Most specific DEGs in lk573 were involved in response to oxidative stress, biosynthetic process, protein localization, and vesicle-mediated transport, and most specific DEPs were assigned to an oxidation-reduction process, carbohydrate metabolic process, and protein phosphorylation. There were 96 genes specifically differentially expressed at both transcriptomic and proteomic levels in lk573. These results revealed the molecular mechanism of salt tolerance and provided candidate genes for further study and salt-tolerant improvement in hulless barley.

## 1. Introduction

Barley (*Hordeum vulgare* L.) is one of the most important cereal crops in the world that is widely used in the brewing industry and for healthy food products [1]. Moreover, hulless barley (*H. vulgare* L. var. *nudum*) is well adapted to extreme environmental conditions and has a long cultivation history as a food crop for Tibetans, mainly cultivated in the Qinghai-Tibet Plateau [2]. Soil salinization is one of the most serious environmental issues. More than 6% of the world’s total land area and 20% of the total agricultural land are affected by salinity [3]. With the degradation of grassland and increased soil salinization in the Qinghai-Tibet Plateau, crop cultivation has also been seriously affected by salt stress. Barley is a relatively salt-tolerant species compared with wheat, rice, and other cereal crops [3]. A good understanding of the response mechanisms to salt stress is critical for crop improvement in salt tolerance. There are numerous reports on the response of barley leaves and roots to salinity [4,5,6]. The inhibitory effects of salinity on barley seed germination have been reported [7,8,9], but there is no report concerning the comprehensive response mechanism to salt stress during the germination stage at the transcriptomic and proteomic levels in hulless barley.

Soil salinity is one of the major factors harmful to agriculture due to its side effects of osmotic stress and ion toxicity on the growth and development of crop plants [10]. Plant salt tolerance at different developmental stages is controlled by different mechanisms, and each stage must be separately studied using special screening procedures [11]. The successful germination of mature seeds in a saline environment is the beginning of salt tolerance in the plant life cycle. Seed germination is considered to be the most critical phase in the life cycle because of its high vulnerability to injury, diseases, and environmental stresses [12]. This process starts with the uptake of water (phase I), followed by a plateau phase (phase II), and terminates with elongation of the embryonic axis (phase III) [13]. Sufficient stores of mRNAs and proteins in mature seeds are imperative for seed germination, showing that germination is prepared during seed maturation, and seedling growth is prepared in the germination stage [12]. Phytohormones are important to seed germination. The balance between abscisic acid (ABA) and gibberellins (GA) critically affects seed dormancy and germination, based on the inhibition of germination by ABA and activation of dormant seed by GA [13]. Auxin is not usually necessary for seed germination but can influence this process when ABA is present [14]. In addition, the universal second messenger calcium (Ca) also regulates seed germination by interacting with the effect of ABA [15].

The inhibition of salinity stress on seed germination results from osmotic stress, oxidative stress, and ion toxicity, shown by decreasing germination rate and extended germination time [16]. Salt stress leads to excessive reactive oxygen species (ROS), which damages proteins, lipids, and nucleic acids or the cellular structure, resulting in oxidative stress [3], and alters the phytohormone balance, with decreases in GA, auxin, and cytokinin and increases in ABA and jasmonates in plant tissues [14]. Under salt stress, the Ca^2+^ concentration increases and triggers the ABA signal and salt overly sensitive (SOS) pathways to decrease the damage from ROS and to regulate sodium ion homeostasis in plants [17,18,19]. The ABA is central to salt stress responses in plants and triggers ROS signals to alleviate the effect of salinity on seed germination [18,19,20]. In the SOS pathway, SOS3 and SCaBP8 perceive Ca^2+^ signals and interact with SOS2, which activates the plasma membrane Na^+^/H^+^ antiporter (SOS1) and vacuolar Na^+^/H^+^ exchanger (NHX) to limit the Na^+^ concentration in cells [17,21]. Moreover, energy production is regulated in response to salt stress [22]. The energy and nutrient substrates are supplied from endosperm for the growth and development of the embryo [12]. Programmed cell death (PCD) occurs in aleurone layer cells to enhance the supply process [12]. Thus, more energy and substrates may be required not only for seed germination but also for resistance to salinity stress.

Recently, the integration of various omics technologies is an effective strategy to promote a better understanding of mechanisms in response to environmental stresses. The molecular mechanism of seed salt response during the germination stage is extremely complex. In the current study, two contrasting hulless barley landraces lk621 and lk573 were used to investigate the molecular mechanism in response to salinity during seed germination by comprehensive transcriptomic and proteomic analyses. The changes in hormone contents under salt stress were also analyzed between lk621 and lk573. Our results provided deeper insights into the molecular mechanisms of salt tolerance during the germination stage and candidate genes for further study and breeding salt-tolerant cultivars.

## 2. Results

### 2.1. Differences of Seed Germination between Two Landraces under Salt Stress

It is well known that salt stress significantly influences seed germination, and this effect differs among varieties [7,10]. In this study, two hulless barley genotypes lk621 and lk573 showed different responses to salt stress. The imbibition of lk621 and lk573 seeds was completed at 4 h under distilled water (control, CK) or 200 mM NaCl solution (salt treatment, T). At 16 h, lk573 seeds could complete germination under both treatments; however, lk621 seeds germinated under CK but not T treatments (Figure 1A–J). The germination rate of lk621 was significantly affected by salt stress, and there was no significant difference in germination rate of lk573 between CK and T (Figure 1K). These results illustrated that lk621 was more sensitive to salt stress than lk573 during the seed germination stage.

The plant hormone contents in seeds were also affected by salt stress. Compared with CK, ABA contents in lk621 and lk573 were increased by T treatment at 16 h (Figure 2A). The GA content in lk621 was decreased by 26.87% and 46.76% after 4 and 16 h of salt stress, respectively. The GA content in lk573 was reduced by 10.19% at 4 h and by 14.52% at 16 h of salt stress (Figure 2B). The auxin contents in both lk621 and lk573 were significantly decreased by salt stress at 4 h and 16 h (Figure 2C). The ratio of GA/ABA significantly decreased by salt stress in both lk621 and lk573 at 4 and 16 h under salt stress (Figure 2D). These results illustrated the side effect of salt stress on plant hormone in sensitive and tolerant genotypes during the germination stage.

### 2.2. Overview of Transcriptomic and Quantitative Proteomic Analyses

A total of 1238.5 million clean reads were obtained after filtering with a number of 1290.8 million raw reads from the 24 samples of lk621 and lk573 under CK and T treatments at 4 and 16 h. Of clean reads, 79.92–84.72% were successfully mapped to the barley genome, and 74.54–77.89% were uniquely mapped (Appendix A). In the proteomic analysis, a total of 2,139,488 spectra were matched to 10,841 peptides, and 6036 proteins were identified (Appendix A). The number of genes specifically expressed in lk621 and lk573 under T treatment at 4 h was 2233 and 1823, respectively, with 335 genes overlapping (Appendix A). At 16 h, 1545 and 374 genes were specifically detected in lk621 and lk573 under T treatment, respectively, with 48 overlapping genes (Appendix A). 

Four pairwise comparisons of transcriptomes and proteomes were made to identify differentially expressed genes (DEGs) and differentially expressed proteins (DEPs) between CK and T treatments in lk621 and lk573 at 4 and 16 h. There were 507 and 400 up- and down-regulated DEGs, respectively, identified in lk621 at 4 h after salt stress, and correspondingly 244 and 1461 in lk573 (Figure 3A). With the extension of salt stress time, more DEGs were identified at 16 h: 3972 and 4814 up- and down-regulated DEGs in lk621, respectively, and correspondingly 2279 and 3908 in lk573 (Figure 3A). At the proteomic level, 143 and 212 DEPs were up-and down-regulated in lk621 by salt stress at 4 h, respectively, and correspondingly 123 and 152 in lk573 (Figure 3B). At 16 h, there were more DEPs identified: 168 and 136 up- and down-regulated in lk621, respectively, and 212 and 269 in lk573 (Figure 3B).

In addition, the specific DEGs and DEPs in lk573 after salt stress were detected, which could be associated with salt tolerance of lk573. Compared with lk621, 1567 DEGs and 171 DEPs were confirmed as specifically expressed in lk573 at 4 h in the T treatment and, correspondingly, 2744 and 328 at 16 h (Appendix A). These results suggested the different responses to salt stress between genotypes at transcriptomic and proteomic levels.

### 2.3. Gene Ontology (GO) and Kyoto Encyclopedia of Genes and Genomes (KEGG) Pathway Analysis of DEGs and DEPs

To gain more insights concerning DEGs and DEPs, GO functional enrichment analysis was also conducted. The GO annotations showed that all enriched DEGs and DEPs were classified into three categories: biological processes, cellular components, and molecular function. In the transcriptomic analysis, 399 and 496 GO terms were searched in lk621 and lk573 after salt stress at 4 h, respectively, among which 32 and 54 corresponding terms were significantly enriched (Appendix A). At 16 h, 845 and 820 GO terms were searched, and 117 and 92 were significantly enriched in lk621 and lk573, respectively (Appendix A). In the proteomic analysis, 261 and 200 GO terms were searched in lk621 and lk573 after salt stress at 4 h, respectively; and correspondingly at 16 h, 261 and 285 GO terms were searched (Appendix A). 

In the transcriptomic analysis, various DEGs were enriched in 53 and 107 KEGG pathways in lk621 after salt stress at 4 and 16 h, respectively, and correspondingly, for lk573, in 77 and 105 KEGG pathways (Appendix A). In lk621, protein processing in endoplasmic reticulum, arginine and proline metabolism, phenylpropanoid biosynthesis, and carotenoid biosynthesis were the significant pathways after salt stress at 4 h, and most DEGs were significantly enriched in pathways of carbon metabolism, plant hormone signal transduction, and biosynthesis of amino acids. In lk573, most DEGs were significantly enriched in phenylpropanoid biosynthesis and plant hormone signal transduction, pathways after salt stress at 4 h, and correspondingly in phenylpropanoid biosynthesis, MAPK signaling pathway-plant, and plant hormone signal transduction pathways at 16 h. In the proteomic analysis, DEPs were enriched in 52 and 49 KEGG pathways in lk621 at 4 and 16 h, respectively, and correspondingly 53 and 55 pathways in lk573 (Appendix A). In lk621, most DEPs were significantly enriched in metabolic pathways at 4 h under salt stress, and correspondingly in terpenoid backbone biosynthesis and fatty acid biosynthesis pathways at 16 h. In lk573, propanoate metabolism and protein export were the significantly enriched pathways at 4 h, and DEPs were significantly enriched in pathways of nitrogen metabolism, pentose and glucuronate interconversions, galactose metabolism, homologous recombination, ascorbate and aldarate metabolism, ABC transporters, folate biosynthesis, and ubiquinone and other terpenoid-quinone biosyntheses. These results revealed that the differences at metabolism levels between salt-sensitive and salt-tolerant genotypes indeed existed in hulless barley.

In addition, GO analysis was also conducted based on the specific DEGs and DEPs in lk573. In the biological process category, most of the specific DEGs were involved in response to stimulus and biosynthetic process at 4 h, and correspondingly response to oxidation-reduction process and metabolic process for most of the specific DEPs (Figure 4A). In the cellular component category at 4 h, most of the specific DEGs were assigned to the cell periphery and cell wall, and most specific DEPs were enriched to membrane and cytoplasm (Figure 4B). For the molecular function category, most specific DEGs were annotated to the GO terms of protein heterodimerization activity and protein dimerization activity at 4 h, and most of the specific DEPs to the GO terms of protein binding and oxidoreductase activity (Figure 4C). The GO terms of specific DEGs and DEPs at 16 h are shown in Appendix A. In the biological process category, most specific DEGs were involved in vesicle-mediated transport and response to oxidative stress, and correspondingly oxidation-reduction process and metabolic process for most specific DEPs. In the cellular component category, most specific DEGs were assigned to macromolecular complex and cytoplasm, and most specific DEPs to nucleolus and membrane. For the molecular function category, most specific DEGs were annotated to the GO terms of heme binding and tetrapyrrole binding, and most specific DEPs to the GO terms of ATP binding and nucleic acid-binding. Finally, the networks of GO terms were obtained using BiNGO to identify the GO terms enriched among these specific DEGs in lk573 (Figure 5 and Appendix A).

### 2.4. Correlation between Transcripts and Proteins

The correlations between transcriptome and proteome profiles were assessed using r values. At 4 and 16 h, r values between expressed transcripts and proteins were 0.0387 (*p* = 0.017) and 0.0316 (*p* = 0.047) for lk621, respectively, and correspondingly –0.0008 (*p* = 0.960) and −0.037 (*p* = 0.023) for lk573 (Appendix A). Between DEGs and DEPs, there was a negative correlation (r = −0.3225, *p* = 0.1) for lk621, and a positive correlation (r = 0.1634, *p* = 0.517) for lk573 at 4 h (Appendix A). At 16 h, the corresponding r values were 0.0254 (*p* = 0.789) and −0.0458 (*p* = 0.651) (Appendix A). These results indicated that the correlation between transcriptome and proteome under salt stress condition was weak during seed germination. The genes with significant differential expression listed in Appendix A were co-expressed at both transcriptomic and proteomic levels. At 4 h, the expression trend of 11 genes in lk621 at the transcriptomic level was consistent with that at the proteome level, while 13 genes were expressed in an opposite fashion, and, in lk573, the corresponding numbers of genes were 13 and five; in lk621, at 16 h, the numbers were 66 and 58; and, in lk573, at 16 h, there were 52 and 50.

### 2.5. Genes Related to Salt Tolerance

Based on the different response of molecular mechanisms to salt stress between lk621 and lk573, 96 genes were specifically differentially expressed in lk573 at both transcriptomic and proteomic levels and are listed in Appendix A. Among these genes, 16 were specifically expressed at 4 h, 80 genes expressed at 16 h, and the gene HORVU6Hr1G066250 expressed both at 4 h and 16 h. These genes might be responsible for salt tolerance of lk573. The relative expression levels of 10 genes selected from among the 96 DEGs at 4 and 16 h were validated by qRT-PCR, and the expression trends were consistent with the expression patterns measured by RNA-seq (Figure 6). Based on the functions of proteins encoded by 27 selected genes (Table 1), a putative model for salt tolerance was constructed for hulless barley (Figure 7). These genes were involved in the balances of energy and substrates supply, cell wall resistance, ion transport, Ca-dependent regulation, phytohormone pathways, ROS reducing, and vesicular trafficking.

During seed germination, many catabolic reactions occur in endosperm cells to provide energy and substrates for seed germination, a large number of proteins involved in energy and substrates metabolism are synthesized or degraded, and PCD occurs in aleurone layer cells [12]. In the putative model (Figure 7), some genes involved in protein metabolism in endosperm cells were specifically regulated at both transcriptomic and proteomic levels, and the proteins involved in PCD of the aleurone layer cells were down-regulated in lk573 after salt stress, which might provide more energy and substrates for seed germination and response to salt stress. The complex mechanism of salt stress response in the embryo cell is shown in Figure 7. First, cell wall resistance plays an important role in lk573 salt tolerance, and some proteins involved in cell wall synthesis and remodeling are specific DEPs in lk573 after salt stress [23]. Then, the NHX and SOS pathways involved in limiting the Na^+^ content in cells and some proteins may influence these pathways to enhance salt tolerance of lk573 [18]. In the process of seed germination during salt stress, Ca^2+^ acts as the messenger, participating in the regulation of the ABA and SOS pathways [24,25]. Moreover, phytohormones play an important role in seed salt tolerance. The ABA pathway is central to salt tolerance by activating ROS reduction, and the GA, auxin, and brassinosteroid (BR) pathways trigger the ABA pathway to influence seed tolerance to salt stress during the germination stage [14,17,26,27]. Finally, some proteins, specifically regulated by salt stress in lk573, that are involved in vesicular trafficking may increase seed salt tolerance by reducing ROS damage [28,29,30].

## 3. Discussion

Seed germination is vulnerable to salinity and varies among different genotypes [9,12]. Two genotypes with contrasting performance under salt stress were used in the current study. Being relatively salt-tolerant, the germination rate was significantly higher for lk573 than lk621 under 200 mM NaCl stress (Figure 1K). As effective strategies, transcriptomic and proteomic analyses were widely used in revealing the molecular mechanism, responding to abiotic stresses [31,32]. To further reveal the molecular mechanism in response to salt stress and to obtain a novel understanding of salt tolerance during seed germination, an integrated analysis of transcriptomic and proteomic levels in lk573 and lk621 was conducted using RNA-seq and iTRAQ technologies. It is reported that salt stress delays phases I and II during seed germination [7,33]. Seedling growth is mainly prepared in phase II [12]. Thus, more biological processes occur in phase II, and seeds should be more vulnerable to salinity in phase II than in phase I. In this study, fewer DEGs and DEPs were detected at 4 than at 16 h (Figure 3), suggesting that seeds were more seriously affected in phase II than in phase I by salt stress during the germination stage. Compared with lk621, phases I and II were not delayed in lk573, suggesting that there were some special mechanisms of lk573 in salt tolerance. It is possible to exploit favorable genes for barley breeding in salt resistance through the integration of transcriptomic and proteomic analyses.

Salt stress triggers a series of responses in plants, including signal transduction, ion transport, and energy and substrate metabolism [17,18,19]. The GO and KEGG pathway analysis showed the complexity of molecular mechanisms in response to salt stress during the seed germination stage. The correlation between the transcriptome and proteome profiles varies with species, response environments, and different stages of growth and development [34,35,36]. In this study, the r values indicated a weak correlation between the transcriptome and proteome profiles under salt stress. Thus, a comprehensive analysis of transcriptome and proteome is necessary to find out the response mechanisms to salt stress during seed germination. We also found some genes were possibly related to salt tolerance by comparing lk573 with lk621 (Appendix A). These genes were involved in energy and substrate metabolism, ion transport, PCD, signal transduction, cell wall stability, phytohormone balance, vesicular trafficking, and ROS reducing. According to co-expression at transcriptomic and proteomic levels, 27 specifically expressed genes in lk573 were focused on, and a putative model for seed salt tolerance was constructed (Figure 7). Thus, some novel insights into salt tolerance during seed germination were obtained.

### 3.1. Energy and Substrates Supplied by Endosperm under Salt Stress

During seed germination, many catabolic reactions occur in endosperm cells to provide energy and substrates for seed germination, and a large number of enzymes involved in metabolism are synthesized or degraded, and PCD occurs in aleurone layer cells [12]. The glycosyltransferase family 61 protein (GT61) has been confirmed to be involved in the synthesis of xylan, one of the main components of the cell wall [37]. In the putative model (Figure 7), GT61 was down-regulated by salt stress, which could result in decreasing the cell wall formation along with the PCD pathway in aleurone layer cells. At the same time, the histone H2A 6 (HTA6) of lk573 was down-regulated in lk573 to match the decrease in chromatin. One tubulin alpha-4 chain (TUA4), the major constituent of microtubules [38], was down-regulated in lk573 under salt stress, which might also occur in the aleurone layer cells with the emergence of PCD. These results suggested that the increase of PCD under salt stress could supply more energy and substrates for seed salt-resistance and germination, and the detection of PCD and energy metabolism would demonstrate this point.

Protein metabolism occurs in endosperm cells during seed germination. Under salt stress, proteins are hydrolyzed into various amino acids, and thus more energy is supplied (Figure 7). The storage protein glutelin type-B-like protein (GLUB2) was significantly down-regulated in lk573 at 16 h after salt stress in this study, showing that protein metabolism was enhanced in response to salt stress. Moreover, the inter-alpha-trypsin inhibitor heavy chain-related (ITIH) and serpin 3 (SPR3) were specifically down-regulated in lk573 under salt stress. ITIH is one component of the inter-alpha-trypsin inhibitor, which was originally discovered in urine and serum due to its inhibitory activity against trypsin [39]. The serpin family is the largest and the most widespread superfamily of protease inhibitors and plays an important role in the process of development and abiotic stress in plants [40], but the definite mechanisms of stress defense are unknown. In this study, the down-regulation of ITIH and SPR3 meant more proteins were degraded, showing their critical roles in energy and substrates metabolism for a response to salt stress. Further research is needed to classify their functions in response to salt stress during seed germination.

### 3.2. Cell Wall in Salt Tolerance

The cell wall is important for protecting plants from biotic stress, and its critical role in abiotic stress is widely discussed [18,31]. The UDP-glucose 4-epimerase (UGE) plays an important role in galactose metabolism and the biosynthesis of galactose-containing polysaccharides and can also participate in arabinose metabolism, affecting cell wall resistance [41,42]. The barley *HvUGE1* gene is orthologous to *AtUGE4* and also necessary to control carbohydrate partitioning in the cell wall [43]. The HvUGE1 protein was only significantly up-regulated by salt stress at 16 h, which suggested that lk573 exhibited a strong salt tolerance by regulating the formation and stability of the cell wall. Barley xylanase inhibitor genes encode endoxylanase inhibitors, which are orthologous to *Triticum aestivum* xylanase inhibitors (TAXI) [44]. It has been confirmed that TAXI-type xylanase is important for the defense against pathogens and wounding [45]. The barley HvXI proteins detected in this research, orthologous to TAXI-IV proteins, might improve salt tolerance of lk573 by maintaining the stability of xylan in the cell wall through inhibiting xylanase activity. The DUF642 (domain of unknown function 642) proteins are also involved in cell wall synthesis [46]. In *Arabidopsis thaliana*, the DUF642 proteins encoded by the *At4g32460* gene can improve seed germination by increasing the activity of pectin methylesterase (PME) [47]. The PME acts an important role in the completion of cell division, involved in some biotic and abiotic stress responses, and is negatively regulated by aldose 1-epimerase (AE) proteins [48,49]. In the current study, the DUF642 protein was exactly up-regulated under salt stress, meaning the enhanced activity of PME to promote seed salt tolerance by regulating the formation of cell walls during the germination stage. The down-regulated AE proteins might coordinate DUF642 protein to increase the PME activity and enhance salt tolerance of lk573 during seed germination. These proteins, involved in the formation and stability of cell wall, were specifically regulated by salt stress in tolerant genotype lk573, suggesting their vital roles of the cell wall in salt resistance during the seed germination stage. 

### 3.3. Ion Transport for Salt Tolerance

Various ionic transporters are involved in ion homeostasis in cells by selective uptake and exclusion of ions [18]. With respect to the NHX and SOS pathways, some Na^+^/H^+^ antiporters, pyrophosphate-energized proton pumps, and H^+^-ATPases were identified in lk621 and lk573 during seed germination. Only one plasma membrane H^+^-ATPase was up-regulated under salt stress in lk621 and lk573 at 4 h, but just at the proteome level, and was unchanged at 16 h at both transcriptome and proteome level. One F-type ATPase was only up-regulated at 4 h, both in lk621 and lk573. Most H^+^-ATPases expressed in lk621 were unchanged and had the same trend as in lk573. The *SOS4* gene encoding a pyridoxal kinase is involved in the biosynthesis of pyridoxal-5-phosphate (PLP), and PLP can modulate the activities of ion transporters to regulate Na^+^ and K^+^ homeostasis [50]. One pyridoxal 5-phosphate synthase subunit (PdxS) was specifically differentially expressed in lk573 at the transcriptomic and proteomic levels. The PdxS protein might influence the expression of SOS4 to regulate ion homeostasis. Furthermore, a ferrochelatase 1 (FC1) was up-regulated specifically in lk573 under salt stress. It has been demonstrated that FC1 can improve salt tolerance by limiting the accumulation of Na^+^ in cells, possibly through the NHX pathway in *Arabidopsis* [51]. FC1 is the terminal enzyme of heme biosynthesis [51]; thus, the heme biosynthesis pathway may coordinate with the NHX pathway to regulate the Na^+^ content in cells during the germination stage.

The NHX and SOS systems have been clearly demonstrated to play important roles in response to salt stress in plant roots and leaves [17,18]. During seed germination, many ionic transporters were not differentially expressed under salt stress in our study. The unchanged expression of these ionic transporters in seeds suggested that the response mechanism to salt stress in seeds is more complicated than in roots and leaves. Ion transport might coordinate other resistant mechanisms to maintain a strong tolerance of lk573 to salt stress during seed germination.

### 3.4. Ca-Dependent Regulation under Salt Stress

The universal second messenger Ca^2+^ triggers the ABA signal and SOS pathways and acts as an important regulator of many processes in plant stress resistance [21,52]. Different abiotic stresses induce Ca^2+^ fluctuations, and the change is decoded by different Ca^2+^-sensing proteins, which contain Ca^2+^-dependent lipid-binding (CaLB) domains or C2 domains [25]. The C2 domain is involved in calcium-dependent phospholipid binding and membrane targeting processes, and it is confirmed to be a CaLB domain [53,54]. Two C2 domain proteins, AtBAP1 and AtCLB, have been confirmed to negatively regulate defense responses in *Arabidopsis* [24,25]. The loss of AtCLB protein function, which is localized in the nucleus of cells and promotes the expression of thalianol synthase gene *AtTHAS1*, has enhanced drought and salt tolerance of *Arabidopsis* [25]. There are also some reports on C2 domain proteins up-regulated by salt stress [21,55]. Thus, the responses of C2 domain proteins to Ca^2+^ fluctuations are various. In our study, a CaLB domain protein (HORVU3Hr1G085130.1) was down-regulated in lk573 after salt stress during seed germination at 4 h. After receiving the Ca^2+^ fluctuation, it might act as a transcriptional repressor to negatively regulate salt tolerance of lk573 during seed germination (Figure 7). However, the downstream protein regulated by CaLB is still unknown, and more details are required to reveal this pathway and demonstrate the critical role of CaLB in seed salt tolerance.

### 3.5. Phytohormones in Salt Stress

It has been well demonstrated that ABA and GA are the primary hormones that antagonistically regulate seed dormancy and germination [13]. Under salt stress, ABA can alleviate the effects of salt stress on seed germination [20]. In this study, the DNA-binding storekeeper protein-related transcriptional regulator, orthologous to the *Arabidopsis* storekeeper-related 1/G-element-binding protein (STKR1), was specifically up-regulated in lk573 at 16 h of salt stress. The STKR1 is involved in the response of plants to environmental stress by interacting with SNF1-related protein kinase 1, the activity of which can be modulated by ABA [56]. So, up-regulated STKR1 may connect to the ABA pathway to regulate seed tolerance of lk573 to salt stress. The lipase/lipoxygenase (PLAT/LH2 family protein) (PLAT) is the downstream target of the ABA signaling pathway and acts as a positive regulator of abiotic stress tolerance [57]. Confusingly, the PLAT protein in our study was down-regulated in lk573 under salt stress. Whether PLAT protein negatively regulates salt tolerance during seed germination needs further investigation.

The role of GA in the regulation of plant responses to abiotic stress has been well discussed, and it is generally concluded that the reduction of GA levels is helpful for restricting plant growth to adapt to several stresses [26]. However, adequate GA levels are necessary for seed germination. Under salt stress, the mechanism of GA balance in seeds to coordinate germination and stress remains unknown. In this research, a tetratricopeptide repeat (TPR)-like (TPR) superfamily protein, homologous to the SPINDLY (SPY) protein, was down-regulated by salt stress in lk573. The SPY has been identified as a negative regulator of abiotic stress, probably by integrating environmental stress signals via GA and cytokinin cross-talk [58]. In the putative model, the down-regulated TPR might involve in GA balance to improve seed germination of lk573 under salt stress. The 2-oxoglutarate (2OG) / Fe(II)-dependent oxygenase (2OG-FeIIO) superfamily proteins catalyze various oxidation reactions of organic substances by using a dioxygen molecule and are involved in a wide range of biological processes in plants, including DNA repair, hormone biosynthesis, and various specialized metabolites [59]. The 2OG-FeIIOs in the putative model might participate in GA biosynthesis or repair DNA damaged by Na^+^ to promote seed germination under salt stress (Figure 7).

Auxin is not necessary for seed germination but influences seed germination through its interaction with the ABA pathway [14]. Under salt stress, auxin negatively regulates seed germination [60]. The flavin-containing monooxygenase (FMO) can oxidize a diverse range of substrates and positively regulate the biosynthesis of auxin [61]. In our study, the FMO and tryptophan-tRNA ligase (TTL) were specifically up-regulated in lk573 at 16 h. The reason may be that, when seed germination is completed, auxin is prepared for seedling growth, and FMOs are involved in its content balance. Tryptophan is an important factor affecting auxin synthesis [62]. The up-regulated expression of TTL could reduce the content of tryptophan, and thus auxin synthesis. These results suggested that there were complex regulatory mechanisms to control auxin content in response to salt stress during seed germination (Figure 7).

The BR plays an important role in promoting seed germination by overcoming inhibition of ABA [63] and can increase plant resistance to such environmental stresses as cold, drought, and salinity [27]. Under salt stress, BR also promotes seed germination in *Arabidopsis thaliana* and *Brassica napus* [64]. A delta (24)-sterol reductase (DSR), homologous to *Arabidopsis* DIMINUTO / DWARF1 (DIM) protein involved in BR synthesis [65], was up-regulated in lk573 under salt stress. We suggested that up-regulated DSR facilitated BR synthesis to improve salt tolerance of barley seeds during the germination stage (Figure 7). The results in the present study just suggested that ABA, GA, auxin, and BR could influence seed tolerance to salt stress. Cytokinins are able to enhance seed germination by alleviating stresses of drought, salinity, and heavy metals [14,66,67]. Jasmonic acid has been confirmed to enhance barley seedling salt tolerance [68]. So, other phytohormones may also participate in regulating seed germination under salt stress. In general, the balance of phytohormones not only plays an important regulatory role in seed germination but also coordinates the response to salt stress at the germination stage. 

### 3.6. ROS Reduction for Salt Resistance

Salt stress leads to superfluous production of ROS in plants, and some enzymes can relieve their damage to cells [3]. Thioredoxin reductase (TrxR) catalyzes the NADPH-dependent reduction of oxidized thioredoxin, which could play a key role in protecting cells from ROS damage [69,70]. In this study, the specific up-regulation of TrxR in lk573 might have increased the activity of oxidized thioredoxin to reduce ROS and enhance seed salt tolerance. An adenine nucleotide alpha hydrolases-like (ANAH) superfamily protein was also up-regulated by salt stress in lk573. This is orthologous to the *Arabidopsis* universal stress proteins (USPs) (At3g53990), and overexpression of USPs can confer a strong tolerance to heat shock and oxidative stress [71]. Confusingly, another ANAH protein (HORVU3Hr1G077000.2) was down-regulated at 16 h after salt stress in lk573, suggesting a diversity of ANAH proteins’ functions in response to salt stress.

Multiprotein bridging factor 1 (MBF1) is a highly conserved transcriptional coactivator with three members in its family: MBF1a, MBF1b, and MBF1c [72]. The MBF1c is specifically elevated under different abiotic stresses, such as salinity, drought, and heat, and may function as a regulatory component between ABA and stress signal pathways [72,73]. In barley, there is still no report on the characterization of the *HvMBF1c* gene. In the putative model, MBF1c might be affected by ABA and activate a gene that participates in reducing ROS levels (Figure 7). In a word, ROS reduction maintained a strong tolerance of lk573 seeds to salt stress during the germination stage.

### 3.7. Vesicular Trafficking in Salt Tolerance

The protein TBC1D15 contains a TBC (Tre-2/Bub2/Cdc16) domain, which can stimulate the intrinsic GTPase activity of Rab7 and functions as the key regulator of intracellular vesicular trafficking [28]. In *Arabidopsis*, Rab7 has been demonstrated to be a positive regulator in tolerance to salt and osmotic stresses by reducing the ROS levels in cells [29]. The exocyst acts as a tethering complex and effector of Rho and Rab GTPases to participate in vesicular trafficking [30]. In the current study, one TBC1D15 protein and one exocyst complex component 1 (ECC1) were specifically up-regulated in lk573 under salt stress (Figure 7). This could improve the GTPase activity of Rab7, and so reduce the damage from ROS to cells. Moreover, one prenylated rab acceptor (PRA1, HORVU3Hr1G088130.1) family protein was up-regulated in lk573 at 4 h under salt stress. The PRA1 proteins also act as receptors of Rab GTPases to regulate vesicle trafficking [74]. In *Arabidopsis*, both overexpression and knockdown of the *PRA1.F4* gene have increased sensitivity to high salt stress and lowered vacuolar Na^+^/K^+^-ATPase and plasma membrane ATPase activities of plants [75]. Two other PRA1 proteins (HORVU3Hr1G018940.1 and HORVU1Hr1G070360.1) were down-regulated at 16 h. Thus, the functions of PRA1 proteins in response to salt stress during seed germination are complex. These results suggested that vesicular trafficking could improve salt tolerance by triggering ROS regulation during seed germination.

We also found that a small subunit processome component homolog (KRR1) was specifically differentially expressed in lk573 under salt stress (Figure 7). The KRR1 is involved in the assembly of the 40S ribosomal subunit [76], which will influence protein translation. However, the mechanism involved in enhancing salt tolerance with this protein during seed germination is unknown, and further identification is needed. The up-regulated KRR1 may increase the expression of genes involved in regulating seed salt responses.

## 4. Materials and Methods 

### 4.1. Seed Germination under Salt Stress

Two hulless barley landraces with purple seeds (lk621 and lk573) collected from Menyuan and Huangyuan countries of Qinghai province, China, respectively, were selected as research objects. Each of the 30 grains of lk621 and lk573 was washed three times with sterile deionized water and then placed in 9-cm Petri dishes containing two sheets of filter paper, moistened by distilled water (control, CK) or 200 mM NaCl solution (salt treatment, T) for 4 and 16 h. Seed morphology was surveyed and photographed by a stereomicroscope (Leica-M165 C; Leica, Wetzlar, Germany). There were three biological replications. Seed germination was determined by the protrusion of the radicle.

### 4.2. Plant Hormone Detection

For content analysis of ABA, GA, and auxin in germinating seeds, 1 g of seeds of each sample were ground in liquid nitrogen. The quantification of these hormones in seeds was performed by indirect ELISA, as previously described [77]. There were three biological replications.

### 4.3. Transcriptome Sequencing Analysis

Total RNA was isolated from 1 g each of seeds of 24 samples [2 genotypes (lk621 and lk573) × 2 treatments (CK and T) × 2 germination time points (4 and 16 h) × 3 biological replications] using the TRIzol reagent (Invitrogen, Carlsbad, CA, USA) according to the manufacturer’s instructions. RNA purity was checked by a NanoPhotometer^®^ spectrophotometer (IMPLEN, Schatzbogen, Munich, Germany). The RNA concentration was measured on a Qubit^®^ 2.0 Fluorimeter (Life Technologies, Carlsbad, CA, USA). The assessment of RNA integrity was performed by the Bioanalyzer 2100 system (Agilent Technologies, Santa Clara, CA, USA). A total amount of 3 µg of RNA per sample was prepared for building sequencing libraries, which were generated using NEBNext^®^ UltraTM RNA Library Prep Kit for Illumina^®^ (NEB, Ipswich, MA, USA), following the manufacturer’s recommendations. The PCR was performed with Phusion High-Fidelity DNA polymerase, and library quality was assessed on an Agilent Bioanalyzer 2100 system. The libraries were sequenced on an Illumina Hiseq 2000 platform (NCBI; BioProject ID: PRJNA578897). Barley genome and gene model annotation files were downloaded from the genome website (http://webblast.ipk-gatersleben.de/barley_ibsc/downloads/). Clean reads were obtained by removing reads containing adapters, reads containing poly-N, and low-quality reads from raw data. At the same time, Q20, Q30, and GC content of the clean data were calculated. Paired-end clean reads were aligned to the reference genome using Hisat2 v2.0.5. The featureCounts v1.5.0-p3 was used to count the read numbers mapped to each gene. Then, FPKM (expected number of fragments per kilobase of transcript sequence per million base pairs sequenced) of each gene was calculated based on gene length and reads count aligned to this gene. Genes with FPKM > 1 were defined as being expressed. Differentially expressed genes (DEGs) were screened between the treated and control groups with three replicates, performed using the DESeq2 R package (1.16.1). Genes with an adjusted *p*-value < 0.05 were assigned as DEGs.

### 4.4. Proteome Analysis

Samples for protein extraction were prepared as for RNA-seq. The extraction of protein was performed with NitroExtraTM (Cat. PEX-001-250ML, N-Cell Technology, Shenzhen, China) using the manufacturer’s instructions. The protein concentration was determined with a Bradford assay. The protein of each sample was digested with trypsin and then desalted with C18 cartridge. Desalted peptides were labeled with iTRAQ reagents (iTRAQ^®^ Reagent-8PLEX Multiplex Kit, Sigma, Foster, CA, USA), as instructed by the manufacturer. The labeled peptide mix was fractionated using a C18 column (Waters BEH C18 4.6 × 250 mm, 5 μm) on a Rigol L3000 HPLC system (Rigol, Beijing, China). The resulting spectra from each fraction were searched against the Hordeum_vulgare_Customer FASTA database containing 81,279 sequences (http://webblast.ipk-gatersleben.de/barley_ibsc/). The raw data were analyzed by Proteome Discoverer software (ver. 2.2, Thermo Fisher Scientific, Waltham, MA, USA) and then probed on the Mascot search engine (ver. 2.3.02; Matrix Science, London, UK). To reduce the probability of false peptide identification, only peptides at a 95% confidence interval (*p* < 0.05), with a false discovery rate estimation ≤5%, were counted as successfully identified. Each positive protein identification contained at least one unique peptide. Proteins containing at least two unique spectra were selected for quantification analysis. Quantitative protein ratios were weighted and normalized by the median ratio in Mascot. Statistical analysis was conducted using Fisher’s test. Differentially expressed proteins (DEPs) with *p* < 0.05, and a > 1.2-fold or <0.83-fold cutoff were considered as up- or down-regulated, respectively. The MS-based proteomics data is available via ProteomeXchange with identifier PXD016100.

### 4.5. Bioinformatics Analysis

Gene ontology (GO) enrichment analysis was implemented by the cluster Profiler R package. The GO terms with corrected *p*-value < 0.05 were considered significantly enriched by DEGs and DEPs. The Kyoto Encyclopedia of Genes and Genomes (KEGG) pathway analysis was performed to investigate high-level functions and utilities of the biological system. The cluster Profiler R package was used to test the statistical enrichment of DEGs and DEPs in KEGG pathways. The significant GO terms enriched among specific DEGs in lk573 were also analyzed via BiNGO to create networks of GO terms [78].

### 4.6. Correlation between Transcript and Protein

According to the fold change of expressed transcripts and proteins between the treated and control groups in lk621 and lk573, the Pearson correlation coefficient (r) was calculated to evaluate the concordance between transcriptome and the proteome profiles [25].

### 4.7. Quantitative RT-PCR 

Quantitative RT-PCR (qRT-PCR) analysis was used to verify RNA-seq results based on 10 selected genes related to salt stress. The primers of these 10 genes are listed in Appendix A. The RNA samples used for qRT-PCR assays were the same as those used for RNA-seq. The qRT-PCR was performed with SYBR^®^ PremixDimerEraser™ (Takara, Dalian, China) according to the manufacturer’s specifications. The reaction mixtures were incubated at 95 °C for 3 min, followed by 40 cycles of 95 °C for 5 s, 60 °C for 60 s, and 72 °C for 30 s. Barley ACTIN gene (AY145451) was used as a control to normalize the amount of gene-specific RT-PCR products. Based on the melting curve analysis of PCR amplicons, the results with specific peaks were selected to assess the expression level of selected genes. The relative expression levels of selected genes were calculated with the 2^−ΔΔ*C*t^ method [79]. According to the expression levels assessed by qRT-PCR and RNA-seq, the correlation was estimated with the fold changes regulated by salt stress.

## 5. Conclusions

Seed germination is sensitive to salt stress, and the mechanism of this process is complex. This is the first report on the research of molecular mechanisms in response to salt stress during the germination stage at transcriptomic and proteomic levels in hulless barley. A large number of genes and proteins associated with salt response were detected. Landrace lk573 was much more tolerant to salt stress than lk621. Moreover, a putative model for expounding salt tolerance of lk573 was constructed (Figure 7), which involved energy and substrate metabolism, cell wall resistance, ion transport, Ca-dependent regulation, phytohormone pathways, reducing ROS levels, and vesicular trafficking. These results deepened our understanding of the mechanism responding to salt stress during seed germination and provided candidate genes for salt-tolerant improvement in hulless barley.

## Figures and Tables

**Figure 1 ijms-21-00359-f001:**
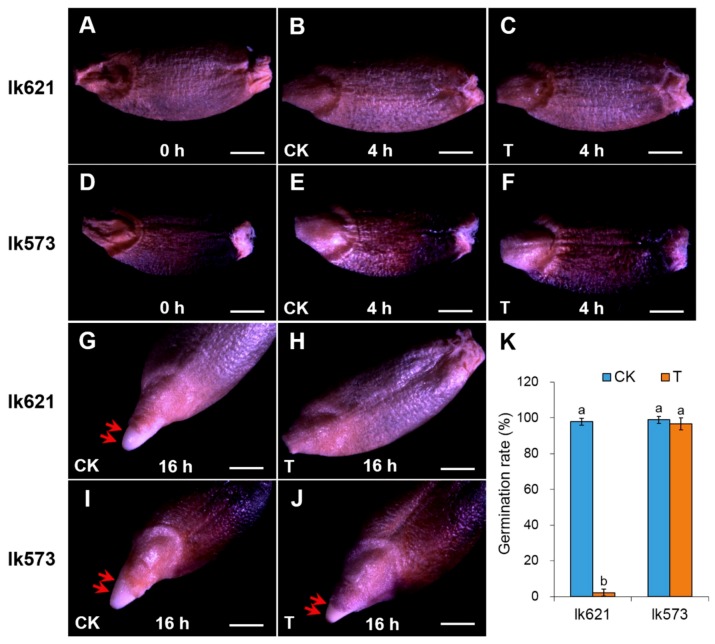
Seed morphology of hulless barley lk621 and lk573 treated with distilled water (control, CK) and 200 mM NaCl solution (salt treatment, T) (**A**–**J**) and seed germination rate (**K**). (**A**,**D**) Dormant seed of lk621 and lk573; (**B**,**C**) lk621 seed under CK and T at 4 h; (**E**,**F**) lk573 seed under CK and T at 4 h; (**G**,**H**) lk621 seed under CK and T at 16 h; and (**I**,**J**) lk573 seed under CK and T at 16 h. Red arrows show the radicles. Bar = 50 μm. Values presented are means of three replicates ± standard error (SE). Different lowercase letters indicate significant difference at *p* < 0.05 as determined by Tukey’s Honestly Significant Difference (HSD) test in each sample.

**Figure 2 ijms-21-00359-f002:**
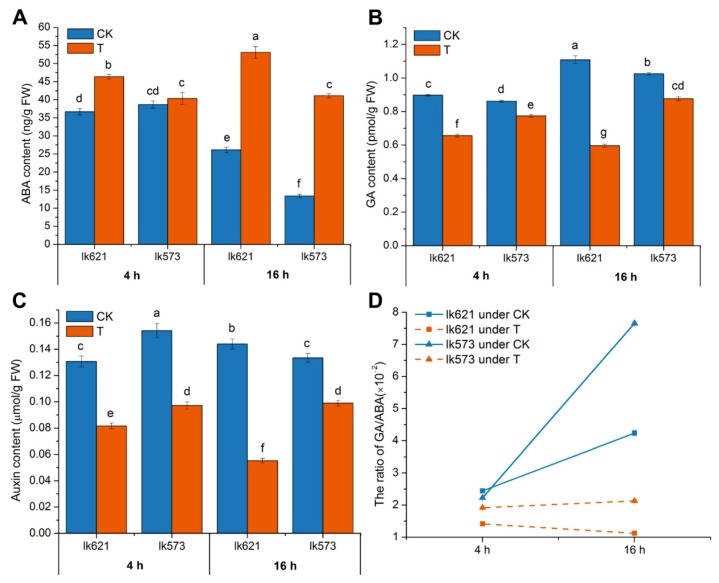
Contents of (**A**) abscisic acid (ABA), (**B**) gibberellins (GA), (**C**) auxin, and (**D**) the ratio of GA/ABA in hulless barley lk621 and lk573 with treatments of distilled water (control, CK) and 200 mM NaCl solution (salt treatment, T) for 4 and 16 h. Values presented are means of three replicates ± standard error (SE). Different lowercase letters indicate significant difference at *p* < 0.05 as determined by Tukey’s HSD test in each sample.

**Figure 3 ijms-21-00359-f003:**
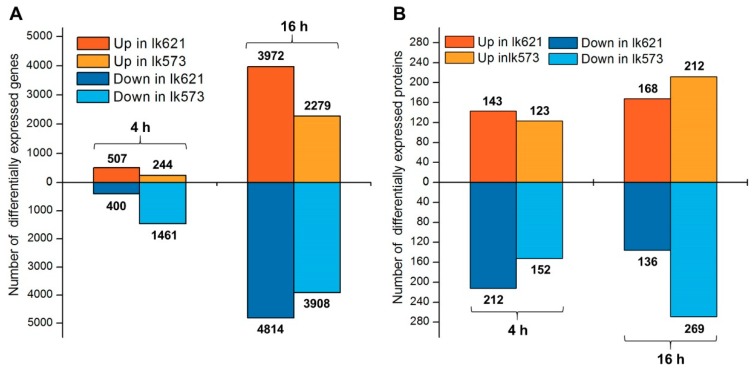
Number of differentially expressed genes (**A**) and differentially expressed proteins (**B**) in hulless barley lk621 and lk573 between treatments of distilled water (control, CK) and 200 mM NaCl solution (salt treatment, T) at 4 and 16 h.

**Figure 4 ijms-21-00359-f004:**
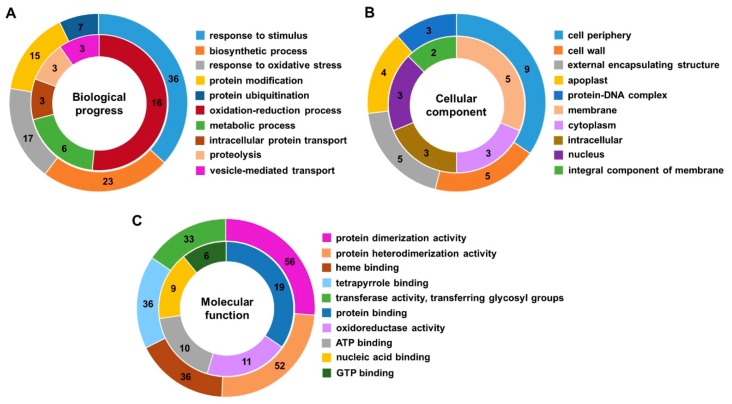
Top five gene ontology (GO) categories: biological process (**A**), cellular component (**B**), and molecular function (**C**) assigned to most of the specific differentially expressed genes (DEGs, outer cycle) and the specific differentially expressed proteins (DEPs, inner cycle) in lk573 at 4 h after salt stress. The numbers represent the number of DEGs or DEPs assigned to each GO term.

**Figure 5 ijms-21-00359-f005:**
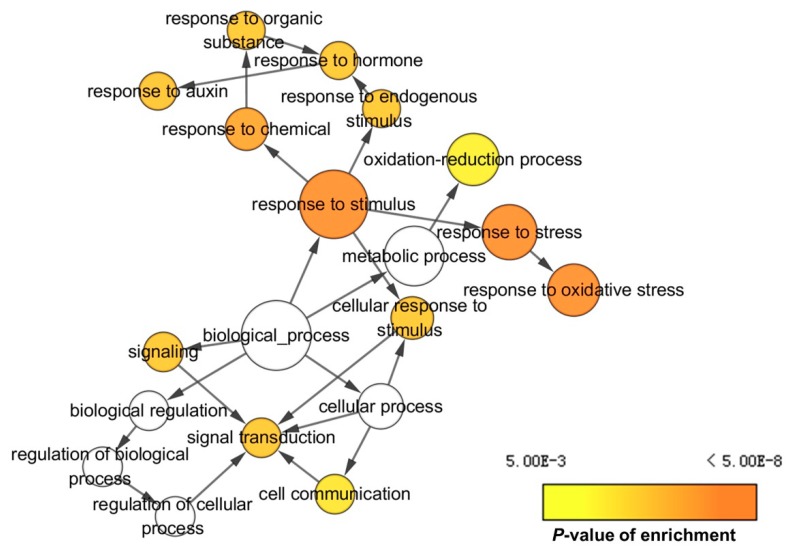
Example of networks representing gene ontology (GO) terms in the biological process category enriched among differentially expressed genes specifically affected by salt stress in hulless barley lk573 at 4 h. Enriched GO terms were identified using BiNGO and visualized with Cytoscape. The GO terms were connected based on their parent-child relationships. Colors of circles indicate the *p*-value of enrichment. Sizes of circles represent the size of GO terms in the background GO annotation. The complete networks of enriched GO terms are presented in Appendix A.

**Figure 6 ijms-21-00359-f006:**
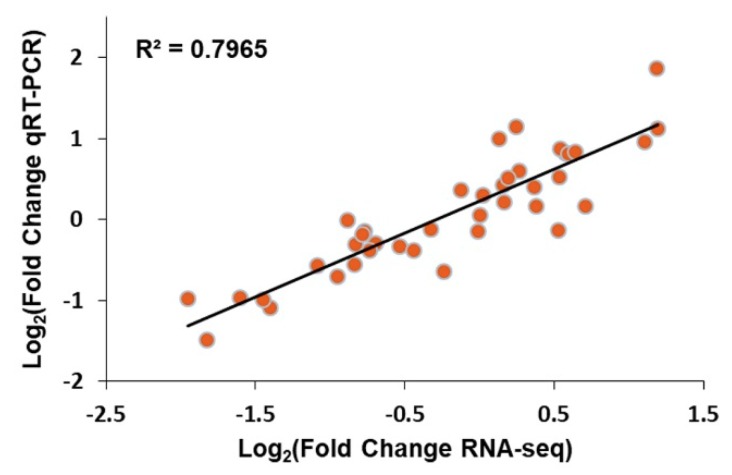
Correlation between RNA-seq and qRT-PCR.

**Figure 7 ijms-21-00359-f007:**
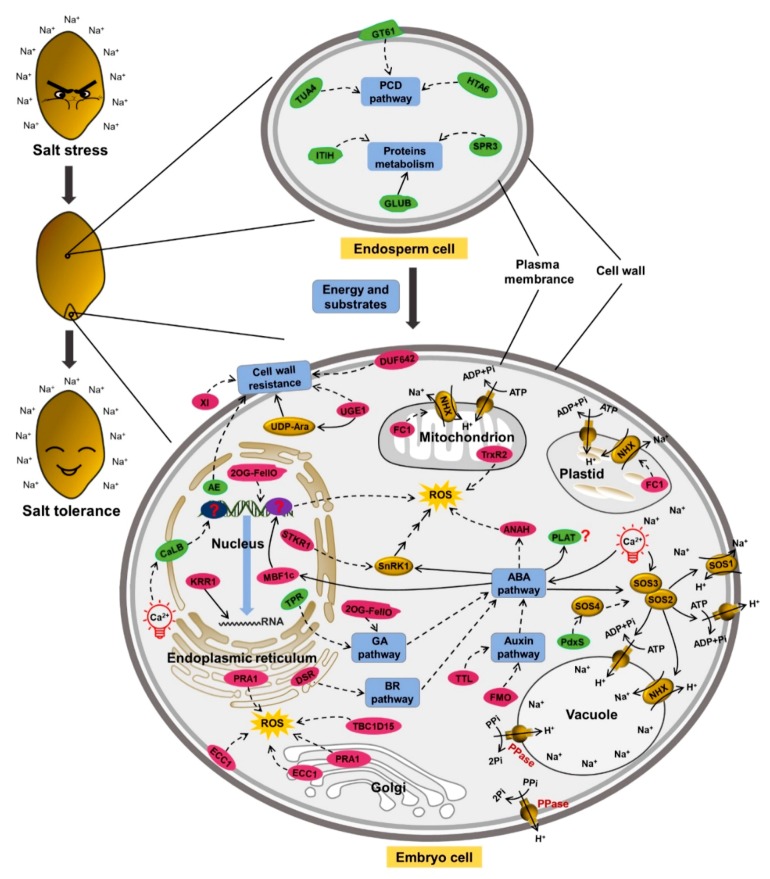
A putative model for salt tolerance of lk573 during seed germination. The enhanced salt tolerance requires energy and substances supplied by endosperm cells, including the critical role of aleurone layer cells. In the embryo, cells need cell wall resistance, reduction in reactive oxygen species (ROS) levels, ionic transport pathway, and different plant hormone pathways, including abscisic acid (ABA), gibberellins (GA), and auxin, along with vesicular trafficking and calcium-dependent regulation. GT61, glycosyltransferase family 61 protein; HTA6, histone H2A 6; TUA4, tubulin alpha-4 chain; ITIH, inter-alpha-trypsin inhibitor heavy chain-related; GLUB2, Glutelin type-B-like protein; SPR3, serpin 3; XI, xylanase inhibitor; DUF642, protein of unknown function DUF642; UGE1, UDP-glucose 4-epimerase 1; AE, aldose 1-epimerase; FC1, ferrochelatase 1; TrxR2, thioredoxin reductase 2; ANAH, adenine nucleotide alpha hydrolases-like superfamily protein; 2OG-FeIIO, 2-oxoglutarate (2OG) and Fe(II)-dependent oxygenase superfamily protein; FMO, flavin-containing monooxygenase family protein; PLAT, lipase/lipooxygenase, PLAT/LH2 family protein; STKR, DNA-binding storekeeper protein-related transcriptional regulator; MBF1c, multiprotein bridging factor 1c; KRR1, KRR1 small subunit processome component homolog; TPR, tetratricopeptide repeat (TPR)-like superfamily protein; DSR, delta(24)-sterol reductase; PRA1, PRA1 (prenylated rab acceptor) family protein; TTL, tryptophan--tRNA ligase; ECC1, exocyst complex component 1; TBC1D15, TBC1 domain family member 15; PdxS, pyridoxal 5-phosphate synthase subunit PdxS; CaLB, calcium-dependent lipid-binding (CaLB domain) family protein; SnRK1, SNF1-related protein kinase 1; UDP-Ara, uridine diphosphate arabinose; PPase, pyrophosphate-energized proton pump; NHX, Na+/H+ exchanger; SOS, salt overly sensitive. The green represents down-regulated differentially expressed proteins (DEPs) by salt stress, and the red represents up-regulated DEPs.

**Table 1 ijms-21-00359-t001:** List of selected genes related to salt tolerance in lk573.

Gene ID	Tran(log_2_FC)	Pro(log_2_FC)	Description
HORVU4Hr1G058810	−1.59814	−0.37389	*Histone H2A 6*
HORVU6Hr1G066250	−0.86759	−0.43371	*Tetratricopeptide repeat (TPR)-like superfamily protein*
HORVU2Hr1G065120	−1.22601	−0.65284	*Pyridoxal 5-phosphate synthase subunit PdxS*
HORVU3Hr1G085130	−0.71579	−0.46692	*Calcium-dependent lipid-binding (CaLB domain) family protein*
HORVU5Hr1G109880	−0.6552	0.28447	*Protein of unknown function, DUF642*
HORVU7Hr1G100810	−0.58782	0.408756	*2-oxoglutarate (2OG) and Fe(II)-dependent oxygenase superfamily protein*
HORVU3Hr1G088130	−1.28183	0.329661	*PRA1 (Prenylated rab acceptor) family protein*
HORVU3Hr1G109590	−0.79428	−0.26857	*Glycosyltransferase family 61 protein*
HORVU5Hr1G098960	−0.83372	−0.38633	*Tubulin alpha-4 chain*
HORVU7Hr1G001030	−1.22656	−0.37861	*Inter-alpha-trypsin inhibitor heavy chain-related*
HORVU5Hr1G087760	−1.60172	−0.48224	*Glutelin type-B-like protein*
HORVU1Hr1G071460	−1.34747	−0.29342	*Serpin 3*
HORVU2Hr1G080100	−0.92463	−0.37249	*Aldose 1-epimerase*
HORVU3Hr1G077000	−0.58028	−0.28372	*Adenine nucleotide alpha hydrolases-like superfamily protein*
HORVU6Hr1G074940	−0.5826	−0.38866	*Lipase/lipooxygenase, PLAT/LH2 family protein*
HORVU5Hr1G067630	−0.82874	−0.90215	*DNA-binding storekeeper protein-related transcriptional regulator*
HORVU3Hr1G100410	−1.02777	0.449302	*Xylanase inhibitor*
HORVU1Hr1G095430	0.561136	0.345589	*UDP-glucose 4-epimerase 1*
HORVU5Hr1G054060	0.639856	0.418968	*Ferrochelatase 1*
HORVU6Hr1G070120	−0.58765	0.266884	*Thioredoxin reductase 2*
HORVU0Hr1G012990	−1.37722	0.861165	*Flavin-containing monooxygenase family protein*
HORVU7Hr1G085130	1.103488	0.418055	*Multiprotein bridging factor 1C*
HORVU3Hr1G067370	0.632487	0.319481	*KRR1 small subunit processome component homolog*
HORVU7Hr1G120030	−0.87435	0.474998	*Delta(24)-sterol reductase*
HORVU5Hr1G025320	0.5301	0.40109	*Tryptophan--tRNA ligase*
HORVU4Hr1G011940	0.657604	0.878678	*Exocyst complex component 1*
HORVU5Hr1G079100	0.495959	0.321981	*TBC1 domain family member 15*

These genes were differentially expressed at transcriptomic and proteomic levels. Tran(log_2_FC), fold change of transcript regulated by salt stress. Pro(log_2_FC), fold change of protein regulated by salt stress. Fold change values are color-coded with red and green for up- and down-regulation, respectively.

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
