# Peer review of "Integrative Transcriptomic and Proteomic Analyses of Molecular Mechanism Responding to Salt Stress during Seed Germination in Hulless Barley"

_ijms, 2020, doi:10.3390/ijms21010359_

Round 1

Reviewer 1 Report

The paper is probably the first work that combines information from transcriptomic and proteomic analyses in order to investigate the molecular mechanisms of salt stress response during the germination stage in Hulless barley (Hordeum vulgare L. var. nudum) that is one of the most important crops in the Qinghai-Tibet Plateau. Soil salinity seriously affects its cultivation. Two contrasting genotypes - lk573 and lk621 - were selected and integrated analysis was performed using RNA-seq and iTRAQ technologies. Compared to the salt-sensitive landrace lk621, the salt-tolerant one lk573 could germinate normally under salt stress. Salt stress delays phases I and II during seed germination. A large number of DEGs and DEPs were identified between control and treatment groups in lk621 and lk573, and fewer DEGs and DEPs were detected at 4 than at 16 h. This suggested that seeds were more seriously affected in phase II than in phase I.
The paper deserves publication after minor revision.
Minor issues:
1. Lines 92-94 are to be removed as they are probably left from the template
2. Line 204 Caption for Figure 5. “differently expressed” -> “differentially expressed”
3. Line 211. P-values are to be provided for these correlations.
4. Line 425. “reductase (TrxR) catalyze” - > “catalyzes”?

Author Response

Response to Reviewer 1 Comments

Point 1: Lines 92-94 are to be removed as they are probably left from the template. 

Response 1: Thank you for your kindly reminder. We apologize for this mistake, and lines 92-94 were already removed in the revised version.

Point 2: Line 204 Caption for Figure 5. “differently expressed” -> “differentially expressed”.

Response 2: We apologize for the error. With your kindly help, the error has been corrected.

Point 3: Line 211. P-values are to be provided for these correlations.

Response 3: Thank you for this very important comment. In the revised version, all P-values have been added for those correlations.

Point 4: Line 425. “reductase (TrxR) catalyze” - > “catalyzes”?

Response 4: Sorry again. With your kindly reminder, we have corrected the error in the revision.

Reviewer 2 Report

The aim of the paper was focused on "Integrative Transcriptomic and Proteomic Analyses of Molecular Mechanism Responding to Salt Stress during Seed Germination in Hulless Barley". The paper is quite interesting, however, some improvements are strongly recommended in order to increase its scientific soundness and contribution to the subject:

1. Authors should place information regarding the quality check of RNA samples (capillary electrophoresis results and RIN number of each sample).

2. I think, some capillary electropherograms of RNA samples evidencing their high purity, integrity and RIN should be added in the Results section or Supplementary file - integrity and purity of RNA samples are crucial in NGS and RT-qPCR studies.

3. Quantitative RT-PCR: Were primers optimized to yield 95%-100% of PCR reaction efficiency?

4. Authors used SYBR Green fluorescent dye during gene expression studies. In this case, it is obligatory to perform Melting Curve Analysis, and results of this examination should be added in the manuscript or Supplementary file (e.g., JPG or TIFF file).

5. Discussion of the results is superficial and should be thoroughly improved.

6. Figure 1 - Duncan’s multiple range tests should be discarded, and the obtained data should be re-analyzed with the use of ANOVA test with the one of the subsequent post-hoc tests.

7. Figure 2 - please, include results of ANOVA test.

Author Response

Response to Reviewer 2 Comments

Point 1: Authors should place information regarding the quality check of RNA samples (capillary electrophoresis results and RIN number of each sample). 

Response 1: Thank you for this very important comment. The quality check of RNA samples was completed before RNA-seq and RT-qPCR. Considering the paper length, the result was not provided in the manuscript. In the revision, we have uploaded a Supplementary file (S2-1 in Supplementary 2) to demonstrate the quality of RNA samples.

Point 2: I think, some capillary electropherograms of RNA samples evidencing their high purity, integrity and RIN should be added in the Results section or Supplementary file - integrity and purity of RNA samples are crucial in NGS and RT-qPCR studies.

Response 2: Yes, we totally agree with your comment. We have uploaded a Supplementary file (S2-1 in Supplementary 2) to show the quality of RNA samples. The details of assessment methods were added in the revised version.

Point 3: Quantitative RT-PCR: Were primers optimized to yield 95%-100% of PCR reaction efficiency?

Response 3: Yes. In this study, more than twenty genes were analyzed by RT-qPCR. After optimizing the PCR system, top ten pairwise primers were selected, and PCR reaction efficiencies of them were all above 95%.

Point 4: Authors used SYBR Green fluorescent dye during gene expression studies. In this case, it is obligatory to perform Melting Curve Analysis, and results of this examination should be added in the manuscript or Supplementary file (e.g., JPG or TIFF file).

Response 4: Thank you for your kindly reminder. We have provided Supplementary figures of Melting Curve (S2-2 in Supplementary 2). In those figures, there were nearly no heterozygous peak. The single peak indicated the specificity of the amplification product.

Point 5: Discussion of the results is superficial and should be thoroughly improved.

Response 5: Thank you for your important suggestion. We have already improved the discussion part thoroughly, please see the details in the revised version. We hope our revise could reach your expect.

Point 6: Figure 1 - Duncan’s multiple range tests should be discarded, and the obtained data should be re-analyzed with the use of ANOVA test with the one of the subsequent post-hoc tests.

Response 6: Thank you for the very important comment. We have already re-analyzed the data by ANOVA test with the Tukey’s HSD test. New Figure 1 was listed in the revised version.

Point 7: Figure 2 - please, include results of ANOVA test.

Response 7: Thank you for your kindly reminder. We also re-analyzed data by ANOVA test with the Tukey’s HSD test, and then the new Figure 2 was also listed in the revised manuscript.

Round 2

Reviewer 2 Report

The Authors partly improved the manuscript, but unfortunatelly, they have confirmed my serious concerns referring to the usage of SYBR Green fluorescent dye during the gene expression studies.

Regarding the Supplementary file 2 (S2-2):

The application of SYBR green dye during qRT-PCR gene quantification is not a precise method. The usage of SYBR Green by the Authors, led to the misleading results that cannot be published at the present form.

Explanation of my concern:
In case of samples no. 2, 4, 6, 7 and 8 occurred non-specific peaks during Melt Curve Analysis. The provided figures strictly evidenced the occurrence of non-specific PCR products (amplicons).

Based on these results, I strongly recommend to re-analyze cDNA samples, using more precise and gene-specific molecular probes (e.g. commonly used TaqMan fluorescent probes).

Author Response

Point 1: The Authors partly improved the manuscript, but unfortunatelly, they have confirmed my serious concerns referring to the usage of SYBR Green fluorescent dye during the gene expression studies. Regarding the Supplementary file 2 (S2-2): The application of SYBR green dye during qRT-PCR gene quantification is not a precise method. The usage of SYBR Green by the Authors, led to the misleading results that cannot be published at the present form. Explanation of my concern: In case of samples no. 2, 4, 6, 7 and 8 occurred non-specific peaks during Melt Curve Analysis. The provided figures strictly evidenced the occurrence of non-specific PCR products (amplicons).

Based on these results, I strongly recommend to re-analyze cDNA samples, using more precise and gene-specific molecular probes (e.g. commonly used TaqMan fluorescent probes).

Response 1:

Thank you very much for this comment. We apologize for the misleading integrated figures in Supplementary file 2 (S2-2). Figures in the Supplementary file 2 (S2-2) were the unsorted figures of Melt Curve Analysis results, thus there were some non-specific PCR products. As you know, in order to show the authenticity and original of the data, we uploaded them as integrated figures which contain specific and non-specific peaks. Actually, in the next step of calculation, the data with non-specific peaks were abandoned. Gene expression analysis in the manuscript was based entirely on the specific amplification results. We have uploaded individual figure for each PCR product in Supplementary file 3 to verify integrated figures in Supplementary file 2 (S2-2). Figures in Supplementary file 3 (S3-1) represented results of un-perfect PCR products, which were discarded in the following analysis of gene expression. Figures in Supplementary file 3 (S3-2) represented the results of specific PCR products.

As you know, SYBR Green fluorescent dye is widely used in gene expression analysis on the premise that there is no-specific amplification in PCR product. Please do not doubt our research results because of the unsorted figures in Supplementary file 2 (S2-2).

Furthermore, we also revised the discussion again. Please see the details in the revision.

Thank you very much again for your efforts and responsible attitude towards improving our manuscript.

We wish you have a happy and great new year 2020!

Round 3

Reviewer 2 Report

I think, the manuscript may be acceptable, after minor revision of English style and grammar.

In addition, the Authors should include explanation in the manuscript, regarding the procedure of processing/transformation of data anf figures obtained after Melting Curve Analysis of PCR amplicons.

Author Response

Response to Reviewer 2 Comments

Point 1: I think, the manuscript may be acceptable, after minor revision of English style and grammar. In addition, the Authors should include explanation in the manuscript, regarding the procedure of processing/transformation of data and figures obtained after Melting Curve Analysis of PCR amplicons.

Response 1:

Thank you very much for your approval and suggestions.

We invited an English native speaker to help the revision of English style and grammar.

According to your suggestion, we have added the explanation of processing data and figures obtained after Melting Curve Analysis of PCR amplicons in the part of “4.7. Quantitative RT-PCR”, i.e. Based on melting curve analysis of PCR amplicons, the results with specific peaks were selected to assess the expression level of selected genes. The relative expression levels of selected genes were calculated with the 2−ΔΔCt method [79]. According to the expression levels assessed by qRT-PCR and RNA-seq, the correlation was estimated with the fold changes regulated by salt stress.

Thank you very much again for all your helps and efforts.